# Precision Approaches in the Management of Colorectal Cancer: Current Evidence and Latest Advancements towards Individualizing the Treatment

**DOI:** 10.3390/cancers12113481

**Published:** 2020-11-23

**Authors:** Rebecca A. Shuford, Ashley L. Cairns, Omeed Moaven

**Affiliations:** 1Department of Surgery, Wake Forest University, Winston-Salem, NC 27157, USA; rshuford@wakehealth.edu (R.A.S.); acairns@wakehealth.edu (A.L.C.); 2Section of Surgical Oncology, Department of Surgery, Mayo Clinic Florida, Jacksonville, FL 32224, USA

**Keywords:** molecular targeted therapies, colorectal cancer, precision oncology, therapy resistance

## Abstract

**Simple Summary:**

Metastatic colorectal cancer (mCRC) represents a significant burden in cancer-related morbidity and mortality. The study of mCRC-related genetic alterations and the molecular landscape of the disease has been ongoing and continues to improve the efficacy of the available treatment options. Here, we review various molecular pathways that are involved in colorectal carcinogenesis with driving mutations that could be targeted for precision approaches in the treatment of mCRC. We summarize groundbreaking clinical trials that are shaping the evolving role of precision approaches in the practice guidelines and discuss the latest advancements in emerging new modalities, novel technologies, and future directions toward individualizing the managemet approaches to the treatment of colorectal cancer.

**Abstract:**

The genetic and molecular underpinnings of metastatic colorectal cancer have been studied for decades, and the applicability of these findings in clinical decision making continues to evolve. Advancements in translating molecular studies have provided a basis for tailoring chemotherapeutic regimens in metastatic colorectal cancer (mCRC) treatment, which have informed multiple practice guidelines. Various genetic and molecular pathways have been identified as clinically significant in the pathogenesis of metastatic colorectal cancer. These include *rat sarcoma* (*RAS*), epithelial growth factor receptor (EGFR), vascular endothelial growth factor VEGF, microsatellite instability, mismatch repair, and *v-raf murine sarcoma viral oncogene homolog b1* (*BRAF*) with established clinical implications. *RAS* mutations and deficiencies in the mismatch repair pathway guide decisions regarding the administration of anti-EGFR-based therapies and immunotherapy, respectively. Furthermore, there are several emerging pathways and therapeutic modalities that have not entered mainstream use in mCRC treatment and are ripe for further investigation. The well-established data in the arena of targeted therapies provide evidence-based support for the use or avoidance of various therapeutic regimens in mCRC treatment, while the emerging pathways and platforms offer a glimpse into the future of transforming a precision approach into a personalized treatment.

## 1. Introduction

Colorectal cancer (CRC) is the third most common malignancy and the fourth leading cause of cancer-related death worldwide [1]. Its incidence has been increasing in recent years as modern lifestyle changes known to increase CRC risk (including dietary changes, obesity, sedentary behaviors, and alcohol and tobacco use) become more widespread, and CRC screening has been widely adopted [2]. These social and epidemiological changes, coupled with improvements in preventive care, sanitation, and communicable disease prevention, have led to an increased age expectancy and an older population with a higher risk of developing CRC. Although adults aged 65 and older are between three and 30 times as likely to be diagnosed with CRC in comparison to younger adults, there is also a trend towards increasingly early onset of CRC in younger individuals [2,3]. Multiple guidelines for the screening, treatment and surveillance of CRC exist. While colonoscopy is the gold standard for screening and diagnosis of CRC, alternatives such as fecal occult blood testing and fecal immunohistochemical testing are also utilized. The United States Preventive Services Task Force (USPSTF) recommends screening asymptomatic adults aged 50 years and older for CRC and the National Comprehensive Cancer Network routinely publishes and updates guidelines for treatment and surveillance of CRC, which are widely used in clinical practice [4,5]. Unfortunately, many barriers prevent patients from receiving appropriate CRC screening. Patient anxiety, out of pocket cost of the procedure, lack of insurance coverage, negative prior clinical experiences have all been linked to under-screening or lack of screening for CRC [6,7,8].

As the number of patients impacted by CRC has climbed, basic/translational research efforts have improved our understanding of the underlying pathophysiology of the disease. The adenoma-carcinoma sequence is a classic model detailing the progressive accumulation of genetic mutations leading from the transition of normal colonic or rectal tissue to an adenomatous polyp, with subsequent inactivation of tumor suppression genes (i.e., *Adenomatous polyposis coli* (*APC), Tumor protein 53 (TP53)*), activation of oncogenes (i.e., *rat sarcoma* (*RAS*)) and eventual development of CRC [1,9]. However, with further advancements, this model has been challenged by the introduction of discrete mechanisms of CRC pathogenesis (which account for approximately 15% cases of sporadic CRC) [9]. For instance, v-raf murine sarcoma viral oncogene homolog b1 (BRAF) mutations and microsatellite instability (MSI) have been implicated in the serrated pathway, in which serrated polyps develop activation of distinct oncogenes [10]. With advancements in our understanding of genetic alterations in CRC, the role of precision medicine is becoming more essential. Only 10% or less of patients have CRC secondary to hereditary syndromes such as familial adenomatous polyposis (FAP) or hereditary nonpolyposis colorectal cancer (HNPCC), which have associations primarily with CRC in addition to other malignancies as well [1,11]. These hereditary CRC syndromes are associated with specific genes, including *MutL homolog 1* (*MLH1*), *MutS homolog 2* (*MSH2*), *APC*, and *mutY DNA glycosylase* (*MUTYH*) [1]. The breadth of genetic involvement in CRC provides an opportunity to expand the evaluation and treatment linked by involved genes, in an era of individualized or precision medicine.

Advancements in translating molecular studies have led to a growing role for personalized treatment for individual patients based on their specific disease process rather than a “one-size-fits-all” approach [12]. Examples of such clinical applications include the use of anti- vascular endothelial growth factor (VEGF) monoclonal antibodies (mAb) (i.e., bevacizumab) in metastatic CRC (mCRC) or anti-epidermal growth factor receptor (EGFR) mAb (i.e., cetuximab) in *panRAS* and *BRAF* wild type (wt) CRC [13]. Multiple guidelines for the screening, treatment, and surveillance of CRC exist. While surgery can be a potentially curative modality even in patients with metastatic disease, comprehensive therapeutic management would entail a multidisciplinary approach, particularly in patients with advanced disease. Precision approaches are of particular interest to improve the efficacy of treatment and achieve better outcomes. Herein, we discuss the commonly involved pathways and actionable therapeutic targets (Table 1).

## 2. Signaling Pathways and Therapeutic Targets

### 2.1. RAS and EGFR

The *rat sarcoma* (*RAS*) genes (including *KRAS* and *NRAS*) are a family of oncogenes that have been linked to the development of CRC in both adenoma-carcinoma sequence and serrated models [10,14]. Overexpression of epidermal growth factor receptor (EGFR) has been identified in 49–80% of CRC, and *RAS* mutations have been identified as a predictor of poor response to anti-EGFR mAb treatment in mCRC [15,16,17,18]. Both the National Comprehensive Cancer Network (NCCN, www.nccn.org) and American Society of Clinical Oncology (ASCO, www.asco.org) guidelines suggest that all mCRC should be tested for extended *RAS* mutation in a certified laboratory and detection of specific mutations would preclude them from anti-EGFR treatment [4,19,20]. It has been suggested that extended *RAS* testing may be helpful in guiding therapy choices. For example, *Neuroblastoma RAS viral oncogene homolog* (*NRAS)* and *Kirsten rat sarcoma viral oncogene homolog* (*KRAS)* exon 2 wt are associated with improved progression-free survival (PFS) and overall survival (OS) with anti-EGFR treatment in addition to chemotherapy [21]. Furthermore, wt status in regard to other biomarkers, including *BRAF/PIK3CA* in addition to *KRAS* and *NRAS* (so called “quadruple negative tumors”), has been demonstrated to have favorable effects in regard to anti-EGFR mAb response [22]. However, in patients with *RAS* wild-type (wt) tumors, the data supports anti-EGFR mAb as an effective adjunct in the treatment of mCRC as described below.

Multiple studies have examined these agents in use with irinotecan-based chemotherapy regimens, as anti-EGFR mAb were postulated to provide benefit to irinotecan-resistant mCRC in human colorectal xenografts [23].

Clinical applications of these findings include the CRYSTAL trial, a phase III study, which showed improved response rate (RR) (RR of 65.2% with the addition of cetuximab compared to 38.6%), PFS (increased by 3 months with the addition of cetuximab; 11.4 months compared to 8.4 months) and median overall survival (OS) (an increase in OS by over 8 months with the addition of cetuximab; 28.4 months compared to 20.2 months) with the first-line treatment of mCRC in patients with RAS wt tumors [24]. The post hoc evaluation of the study extending *RAS* mutation showed that there may be benefits to the addition of cetuximab if *RAS* mutation signals were less than 5% [24]. The phase III EPIC trial showed that the combination of irinotecan and cetuximab in comparison to irinotecan alone improved quality of life, increased PFS (median PFS increased by 1.4 months; 4 months with the addition of cetuximab compared to 2.6 months) and RR (RR 16.4% with the addition of cetuximab, compared to RR of 4.2%), but no difference in OS (10.7 months with cetuximab compared to 10.0 months) in patients who previously failed first-line systemic treatment of mCRC [25]. Another phase III study, the BOND trial, showed improved RR (RR was 12.1% higher with combination therapy; 22.9% versus 10.8%) and time to progression (time to progression was 2.6 months longer with combination therapy; 4.1 months versus 1.5 months), with a trend towards increased median survival (8.6 months with combination therapy versus 6.9 months, not a statistically significant finding) in patients treated with combination cetuximab and irinotecan compared to cetuximab monotherapy [26]. In regard to targeted anti-EGFR treatment in use with oxaliplatin-based regimens, a phase III trial carried out by Douillard et al. showed improved PFS (PFS increased by 2.2 months with the addition of panitumumab; 10.1 months versus 7.9 months) and OS (OS increased by 5.8 months with the addition of panitumumab; 26 months compared to 20.2 months) with FOLFOX and panitumumab in comparison to FOLFOX alone after a post hoc evaluation with extending *KRAS* mutations [27,28]. The phase II OPUS trial compared treatment of mCRC with FOLFOX-4 versus FOLFOX-4 plus cetuximab and found improved PFS (hazard ratio (HR) 0.567, *p* = 0.0064); however, there was no significant effect on OS (HR 1.015, 95% CI 0.791–1.303, *p* = 0.91) [29]. These findings were echoed in the phase III PRIME trial, in which patients were treated with either FOLFOX-4 or FOLFOX-4 plus panitumumab and investigators found improvements in PFS and a trend towards improvement in OS in a subset of *KRAS* wt patients only (median PFS 10.0 months versus 8.6 months, *p* = 0.01 and median OS 23.9 months versus 19.7 months, *p* = 0.17) [28]. Similarly, the TAILOR trial, also a phase III study, demonstrated improved RR (RR increased by 21.6% with the addition of cetuximab; 61.1% versus 39.5%), PFS (improved by 1.8 months with the addition of cetuximab; 9.2 months compared to 7.4 months) and OS (OS increased by 2.9 months with the addition of cetuximab; 20.7 months compared to 17.8 months) with FOLFOX plus cetuximab versus FOLFOX alone in patients with *RAS* and *BRAF* wt disease [30].

Different clinical variables can determine the appropriateness of anti-EGFR treatment. The benefits of this modality differ in regard to use in the neoadjuvant setting of resectable versus unresectable mCRC as well as primary tumor laterality. Anti-EGFR mAbs have benefits in unresectable metachronous CRC metastases (in combination with FOLFIRI or irinotecan) in patients who are proven to have *RAS* wt disease and these data have informed the NCCN guidelines [4,14,31,32,33]. In contrast, the New EPOC trial showed worse outcomes in mCRC patients who were *KRAS* wt, including a significantly shorter PFS (PFS 14.1 months with the addition of cetuximab compared to 20.5 months) and a trend towards worse median OS (median OS was 39.1 months with the addition of cetuximab, median OS was not reached with chemotherapy alone, but this finding was not statistically significant) in mCRC patients with resectable hepatic metastases treated with chemotherapy and cetuximab compared to chemotherapy alone in the neoadjuvant setting [34]. One potential bias of this study is that only *KRAS* wt (and not all *RAS* wt) patients were included. Furthermore, findings of the CAPRI trial demonstrated improved progression-free survival (PFS) in *RAS* wt mCRC patients treated with anti-EGFR mAb by almost 2 months (PFS of 6.4 months in patients who were treated with cetuximab and FOLFOX compared to 4.5 months with FOLFOX alone) [32].

Furthermore, while resistance to targeted EGFR mAb is well documented in cases of *RAS* mutations, several other biomarkers are associated with EGFR resistance. Specifically, *BRAFV600E*, *MET, MEK, PIK3CA, PTEN* and *HER2* are linked to innate EGFR resistance in mCRC [35]. There are also data that show that resistance to EGFR treatment can be acquired de novo [22,36]. Acquired EGFR resistance continues to be a topic of investigation, and SHP2 has been studied as a target in patients with EGFR resistance in non-small cell lung cancer (NSCLC) [37]. Other studies have identified various targets *DUSP4, ETV5, GNB5, NT5E, PHLDA* as markers to overcome EGFR resistance [38]. MEK inhibitors, MET inhibitors, *KRAS* mutant inhibitors and other drugs have been investigated as potential options to combat anti-EGFR resistance [35,39,40]. Liquid biopsy can be used to detect developing EGFR resistance before it is clinically or radiographically evident [40,41]. Ongoing studies continue to elucidate EGFR resistance and the role of biomarkers in this phenomenon.

Mounting evidence suggests that left-sided primary CRC benefits most from treatment with anti-EGFR mAb. In a review of the literature, Sandhu and colleagues found that mCRC patients with *RAS* wt left-sided lesions, in particular, have improved RR (RR ranging 66.4–72.5% with the addition of anti-EGFR mAb treatment compared to RR ranging 40.6–52.6% with chemotherapy alone), PFS (PFS ranging 9.2–12.9 months with the addition of anti-EGFR mAb treatment compared to PFS ranging 7.6–9.2 months with chemotherapy alone), and OS (OS ranging 22–30.3 months with the addition of anti-EGFR mAb treatment compared to OS ranging 18.7–23.6 months with chemotherapy alone) with anti-EGFR mAb treatment [33]. Similarly, the CALGB/SWOG 80405 trial showed overall improved outcomes, especially in left-sided primary *KRAS* wt CRC with improved OS (OS for all-sided lesions was 30 months with cetuximab plus chemotherapy versus 29 months with bevacizumab plus chemotherapy, the finding was not statistically significant) in this population [42,43]. Conversely, PFS (PFS after treatment with cetuximab for left-sided lesions was 12 months versus 7.7 months for right-sided lesions) and OS (OS after treatment with cetuximab for left-sided lesions was 37.5 months versus 16.4 months for right-sided lesions) were shorter in *KRAS* wt patients with right-sided primary CRC who were treated with cetuximab [43]. Interestingly, neither PFS nor OS was improved with the addition of cetuximab compared to bevacizumab without controlling for *KRAS* wt status [43].

While *KRAS* mutations play an important role in decision making for other targeted therapies, *KRAS* inhibitors have not yet been well established for targeted therapy. Hong et al. examined the use of sotorasib (a *KRAS* p.G12C mutant inhibitor) in patients with advanced non-small cell lung cancer (NSCLC), CRC and other solid tumors and reported improved objective response rate (ORR) and disease control (defined as at least partial RR or stable disease) [39]. While their findings were most remarkable in the NSCLC cohort, the CRC cohort did have a disease control rate of 73.8% (95% CI 57.96–86.14%) and a median PFS of 4 months (range 0–11.1 months) [39]. This study is important, as it revealed a clinical benefit in a prospective study at this scale for the first time. While *KRAS* G12C is not a common mutation in colorectal cancer, this study is significant in demonstrating treatment efficacy for *KRAS* inhibitors. Other *KRAS* inhibitors are under investigation in clinical trials.

### 2.2. Microsatellite Instability (MSI), Mismatch Repair (MMR), and Immune Checkpoint Inhibitors

Deficiency in MMR genes (dMMR) is a known risk factor for CRC as a result of multiple successions of genetic mutations. In this pathway, CRC develops as a result of the inactivation of DNA mismatch repair genes, including *MSH2*, *MSH6*, and *MLH1*, which leads to the accumulation of multiple mutations in long repetitive sequences of short DNA fragments (termed microsatellite), and subsequent microsatellite instability [44]. Microsatellite instability (MSI-H) is detected in up to 17% of sporadic CRC, and both MSI-H and MMR are associated with HNPCC [44]. There is heterogeneity within the CRC associated with dMMR and MSI-H, and these differences have been investigated and divided into four subgroups (CMS1, CMS2, CMS3, and CMS4) based on molecular features [45]. MSI-H, which is associated with CRC, confers better prognosis and is associated with a proximal location within the colon, mucinous or signet ring histopathology, and less likelihood of invasion [44]. Furthermore, MSI-H and dMMR have implications for CRC treatment, particularly in the adjuvant setting [44,46]. Further analysis of outcome data from the QUASAR trial by Hutchins et al. demonstrated improved prognosis and lower risk of recurrence of CRC in patients with dMMR, with an 11% recurrence rate in dMMR, compared to a 26% recurrence rate in patients with no mutations or deficiencies in MMR; nonetheless, it was not predictive of chemotherapy response [47,48]. Data from the NSABP C-07 trial support these conclusions and found that dMMR status had a favorable prognosis for recurrence (hazard ratio (HR) 0.48, 95% CI 0.33–0.70, *p* < 0.0001) and improved OS (HR 0.63, 95% CI 0.46–0.89, *p* = 0.0084) in CRC, although there was also demonstrable worse survival after recurrence in patients with dMMR (HR 1.60, 95% CI 1.07–2.41, *p* = 0.02) [49]. Data from the NSABP C-07 trial were not able to predict responses to systemic oxaliplatin [49].

Sargent et al. investigated five randomized control trials in which stage II or stage III CRC patients received surgery with or without adjuvant fluorouracil (FU)-based chemotherapy postoperatively [50]. Patients without MMR mutations or deficiencies had higher disease-free survival (DFS) after adjuvant treatment with FU-based adjuvant therapy in comparison to dMMR patients (HR 0.67, 95% CI 0.48–0.93, *p* = 0.02), and these were statistically significant findings [50]. Interestingly, bevacizumab, an anti-VEGF mAb, was shown to have a survival benefit when used in combination with standard therapy for the treatment of CRC in patients with known dMMR (HR 0.52, 95% CI = 0.29–0.94, *p* = 0.02) [51]. Additionally, MMR mutations are known to affect immune checkpoint proteins PD1, PDL1, CTLA-4, LAG-3, and IDO and these have been investigated as possible immunotherapy targets [52].

Multiple immune checkpoint markers, including PD1, are of great interest as drug targets in melanoma, renal cell carcinoma, small and non-small cell lung cancer, Hodgkin lymphoma, bladder cancer and other malignancies [53]. Currently, multiple phase II studies are showing promising results for the use of anti-PD1 mAb in treatment of mCRC and the NCCN currently recommends treatment of unresectable metachronous CRC metastases with pembrolizumab or nivolumab in patients with dMMR or MSI-H tumors [4]. The Checkmate-142 trial, a phase II study that examined the use of nivolumab in 74 mCRC patients with dMMR or MSI-H, showed a 31.1% overall response rate, with 34.8% having a response for over a year and 68.9% had control of the disease for at least 12 weeks [54]. Le et al. found that treatment with pembrolizumab in patients with dMMR resulted in a RR of 40% (versus 0%) and immune-related PFS of 78% (versus 11%) in comparison to those without MMR deficiency [55]. The KEYNOTE-164 trial, and Furmet et al. investigated the use of other checkpoint inhibitors (anti-PDL1 and anti-CTLA4) [56,57]. The KEYNOTE-164 trial examined the use of pembrolizumab in adults with dMMR or MSI-H mCRC who were previously treated with at least two (cohort A) or one (cohort B) other forms of standard therapy [56]. ORR was 33% in both cohorts, the median PFS was 2.3 months (cohort A) versus 4.1 months (cohort B), while median OS was 31.4 months (95% CI 21.4 months not reached in cohort A) and not reached in cohort B (95% CI 19.2 months not reached), and thus demonstrating more durable responses in less pretreated patients [56]. KEYNOTE-177 is an important phase 3 multicenter, randomized controlled trial that investigated the use of first-line immune checkpoint blockade with pembrolizumab (versus investigator’s choice of standard chemotherapy) in patients with dMMR mCRC [58]. Investigators found a statistically significant improvement in PFS with pembrolizumab compared to chemotherapy (median PFS 16.5 compared to 8.2 months, HR 0.60, 95% CI 0.45–0.80, *p* = 0.0002) [58]. Awaiting publication of the final KEYNOTE-177 results, the U.S. Food and Drug Administration (FDA) has approved pembrolizumab for first-line use in dMMR/MSI-H mCRC based on these findings [59]. Collectively, these data support testing for dMMR in patients with CRC and demonstrate the variable efficacy of systemic treatment, including standard chemotherapy targeted anti-PD1 agents, in selected dMMR patients. Recent and ongoing trials investigating the use of immune checkpoint inhibitors in the treatment of CRC are summarized in Table 2 and Table 3 [53,54,55,56,59,60,61,62].

Despite these findings, several challenges remain in the realm of clinical applications for immune checkpoint inhibitors in mCRC treatment, particularly in microsatellite stable disease (MSS). It is postulated that MSS CRC has an inherent resistance to immune checkpoint blockade, and several investigators have outlined the difficulties in applying immune checkpoint inhibitor therapy to a wider population of mCRC patients with MSS disease secondary to intrinsic resistance of MSS disease to immune checkpoint blockade [62,63]. Several toxicities and adverse events are associated with immune checkpoint inhibitors. These side effects are often immune-related adverse events (irAE) and inflammatory reactions in different organs and systems (“itis”), including gastrointestinal tract, respiratory, endocrine, skin and musculoskeletal irAEs. Manifestations include fatigue, diarrhea, rash, elevated lipase/amylase, hepatitis, diabetes, pneumonitis and colitis [54]. While the future of immune checkpoint inhibitors remains promising for patients with dMMR/MSI mCRC, the search for therapeutic immune checkpoint options and finding methods to convert immunologically “cold” tumors to “hot” tumors and responders to immunotherapy in patients with proficient MMR and microsatellite stable mCRC remains ongoing.

### 2.3. BRAF

V-raf murine sarcoma viral oncogene homolog b1 (*BRAF*) is a proto-oncogene involved in the MAPK signaling cascade and is a downstream effector of EGFR (along with *RAS*) [64,65,66]. Interestingly, inhibition of *BRAF* has been associated with over-activation of EGFR, which is postulated to cause the attenuated response of vemurafenib observed in in-vitro mCRC [46]. *BRAF* V600E activating mutation has been implicated in the development of CRC and is present in approximately 15% of cases of sporadic CRC, and portends a worse prognosis [1,14,64]. *BRAF* has been investigated and developed as a drug target (i.e., vemurafenib) and has been used with success in the treatment of melanoma [14]. The NSABP C-07 trial showed that *BRAF* mutations in CRC were also associated with dMMR, advanced age, a trend towards higher T stage, and decreased survival after recurrence (HR 2.31, 95% CI 1.83–2.95, *p* < 0.0001), and worse OS (HR 1.46, 95% CI 1.20–1.79, *p* = 0.0002) [49]. Similar to the NSABP C-07 results regarding MMR status and response to adjuvant therapy, *BRAF* mutations were not a significant predictor in response to oxaliplatin [49]. Of note, it is well established that *BRAF* mutations clinically translate to lack of response to anti-EGFR mAb in *KRAS* wt mCRC patients and the NCCN recommends *BRAF* testing for all patients with mCRC to assess appropriate candidates for anti-EGFR mAb treatment in this population [4,66,67,68].

These findings have led to studies investigating the efficacy of combining anti-*BRAF* and anti-EGFR agents as a means of circumventing poor response to existing therapeutic regimens. A pilot study by Yaeger et al. showed that anti-*BRAF* therapy (vemurafenib) was tolerated well when combined with panitumumab in *BRAF* V600E mutant mCRC patients who had disease progression on standard therapy [69]. A phase 1B study by Hong and colleagues similarly showed that a combined regimen of vemurafenib, irinotecan and cetuximab was well tolerated in mCRC patients with *BRAF* mutations [70]. The recent S1406 phase II trial investigating the efficacy of combined cetuximab, irinotecan, and vemurafenib compared to cetuximab and irinotecan alone has completed accrual and preliminary results showed an increased PFS (4.3 months with the addition of vemurafenib versus 2.0 months with cetuximab and irinotecan alone); nonetheless, these findings were not statistically significant [71]. Addition of encorafenib (a *BRAF* inhibitor) with or without binimetinib (a MEK inhibitor, with effectors downstream of *BRAF*) in addition to cetuximab significantly improved OS (OS 9 months with triplet therapy versus 5.4 months) in comparison to treatment with cetuximab plus chemotherapy in the phase III BEACON trial [72]. Updated results of the BEACON trial demonstrate similar OS with either the triplet or doublet chemotherapy combinations (median OS 9.3 months, 95% CI 8.2–10.8; versus 9.3 months, 95% CI 8.0–11.3) for triplet and doublet chemotherapy, respectively, and with higher rates of adverse events in the triplet therapy group compared to doublet therapy (65.8% versus 57.4%) [73]. While the findings regarding V600E mutant mCRC have clinical implications for anti-*BRAF* therapy, non-V600E *BRAF* mutations in mCRC patients have been demonstrated to predict poor response to anti-EGFR therapy [74,75]. Furthermore, V600E mutant CRC has been associated with older age, right-sided laterality, poorly differentiated status and advanced disease stage vs. non-V600L mutant CRC, which is associated with left-sided primary tumor location and well-differentiated histology [74].

### 2.4. VEGF

Vascular endothelial growth factor (VEGF) is known to promote angiogenesis, leading to increased tumor growth secondary to more robust vascular supply [76]. Bevacizumab, an anti-VEGF mAb, is currently part of an NCCN recommended first-line regimen for unresectable metachronous mCRC (in combination with standard chemotherapeutic regimens) [4].

The addition of bevacizumab was shown to increase RR (RR 44.8% compared to 34.8%), response duration (median duration of response 10.4 months compared to 7.1 months), PFS (median duration of PFS 10.6 months compared to 6.2 months) and OS (median duration of survival 20.3 months with the addition of bevacizumab compared to 15.6 months) when used in combination with systemic chemotherapy patients with mCRC in the phase III AVF2107 trial [77]. These findings were replicated in the phase III ARTIST trial, which showed significantly increased objective RR (objective RR 35.3% with the addition of bevacizumab compared to 17.2%), PFS (median PFS 8.3 months versus 4.2 months) and OS (OS 18.7 with the addition of bevacizumab compared to 13.4 months) when bevacizumab was added to irinotecan, leucovorin, and 5-FU as first-line therapy [78]. Another phase III study, the TRIBE trial, demonstrated the efficacy of bevacizumab with FOLFIXIRI with a demonstrably higher early tumor shrinkage rate, RR (objective RR 65% versus 53%) and PFS (PFS 12.1 months versus 9.7 months) in the experimental group treated with FOLFOXIRI plus bevacizumab compared to the control group treated with bevacizumab plus FOLFIRI [79,80]. The BRiTE study, a phase IV prospective cohort study, also supported the use of bevacizumab as a first-line adjunct in mCRC treatment [81]. The addition of bevacizumab to FOLFOX4 improved median duration of survival (12.9 months versus 10.8 months, *p* = 0.0011) and 1 year survival (56% versus 43%, *p* < 0.0001) [82]. It should be noted that there are also data on mCRC treated with oxaliplatin-based chemotherapy plus bevacizumab as a first-line regimen, which show that PFS may be significantly improved, and OS may not be significantly improved [83]. These findings and the observations from similar investigations have solidified the role of anti-VEGF mAb as part of a first-line regimen for the treatment of unresectable mCRC. Toxicity is an important adverse clinical outcome, which has also been well described in the literature. The BEAT study assessed the safety of bevacizumab with the physician’s choice of standard chemotherapy and found the most common grade three or greater adverse events were neutropenia, neuropathy, bleeding, gastrointestinal perforation, hypertension, thromboembolic events, proteinuria and poor wound healing [84,85]. Furthermore, genetic analysis of polymorphisms in ERCC1, XPD and GSTP1 did not show any effect on the safety or efficacy of bevacizumab in combination with standard chemotherapy for mCRC [85].

Although bevacizumab is the most widely investigated anti-VEGF agent and is currently recommended by the NCCN as the preferred anti-VEGF mAb considering its more favorable cost and side-effect profile, other anti-VEGF agents, including aflibercept (an anti-VEGF fusion protein) and ramucirumab (an anti-VEGFR2 mAb) have been studied and are acceptable alternatives in the first-line treatment of unresectable metachronous CRC metastases [4]. These alternative anti-VEGF agents have also been studied as the next-line adjuncts after treatment failure. The VELOUR trial, a phase III study, showed improvement in RR (RR 19.8% with the addition of aflibercept compared to 11.1%), PFS (median PFS 6.9 months with the addition of aflibercept compared to 4.7 months), and OS (median OS 13.5 months with the addition of aflibercept compared to 12.1 months) in mCRC patients who had disease progression on or after treatment with standard therapy when treated with aflibercept and FOLFIRI in comparison to FOLFIRI alone as second-line treatment [86]. Similarly, the phase III RAISE trial showed improved overall survival in mCRC patients with disease progression on first-line therapy who received ramucirumab plus FOLFIRI in comparison to FOLFIRI alone as second-line treatment [87]. The success of these newer anti-angiogenesis agents in the second-line setting has set the stage to assess their role as potential first-line therapy options in the future [87,88,89].

The multi-kinase inhibitor regorafenib, with activity against VEGF, now has FDA approval for mCRC, which has progressed with standard treatment. In the phase III CORRECT trial on regorafenib, median OS was improved with regorafenib monotherapy compared to placebo (6.4 months versus 5.0 months, *p* = 0.0052) [89]. Interestingly, in the IMblaze370 study, median OS did not significantly differ between mCRC patients treated with atezolizumab plus cobimetinib or atezolizumab monotherapy versus regorafenib as third-line therapy [90]. Clinical application of anti-VEGF in mCRC remains a challenging obstacle due to a lack of clinically significant biomarkers, which has limited its application with regard to treatment. A recent study has identified Tie2, a tumor vasculature marker, as a relevant clinical marker to monitor anti-VEGF use [91]. In comparing use of standard chemotherapy combined with either bevacizumab or cetuximab in the FIRE-3 trial, there was no significant difference in objective RR (odds ratio 1.18, 95% CI 0.85–1·64, *p* = 0.18) or median PFS (HR 1.06, 95% CI 0.88–1.26, *p* = 0.55) but there was a significantly higher median OS in the cetuximab group compared to bevacizumab (28.7 versus 25.0 months, *p* = 0.017) [92]. A similar study by Venook et al. also studied the addition of cetuximab versus bevacizumab in chemotherapy, and while there was a trend towards improved OS with cetuximab, this finding was not statistically significant (30.0 months versus 29.0 months *p* = 0.08) in *KRAS* wt patients [42].

### 2.5. HER2

Human epidermal growth factor receptor 2 (HER2) plays a role as an oncogene in several solid organ cancers, including breast, lung and ovarian cancer, and has been successfully used as a therapeutic target (i.e., trastuzumab). While studies have failed to support any prognostic value for overexpression of HER2 in CRC, it is present in up 14–81% of CRC cases and HER2 overexpression has also been demonstrated to be predictive of resistance to anti-EGFR treatment [93,94]. The recent phase IIa MyPathway study showed a RR of 32% in refractory mCRC patients with HER2 amplification treated with dual anti-HER2 agents trastuzumab and pertuzumab [95]. Similarly, the HERACLES trial, a phase II study, similarly examined dual anti-HER2 treatment with trastuzumab and lapatinib in *KRAS* wt mCRC patients with HER2 overamplification who were refractory to prior therapies and found a RR of 30% [96]. The HERACLES-B trial expanded on these findings and examined the use of trastuzumab-emtansine in patients with *RAS/BRAF* wt, HER2 amplified mCRC with the progression of disease after treatment with anti-EGFR therapy and found objective RR of 10% (95% CI 0–28%) and median PFS of 4.8 months (95% CI 3.6–5.8 months) [97]. The current HERACLES-RESCUE trial is examining the use of trastuzumab-emtansine in patients with HER2 amplified mCRC, which has progressed with lapatinib and trastuzumab treatment [98]. Similarly, the DESTINY CRC01 trial is examining the use of trastuzumab-deruxtecan in patients with HER2 amplified mCRC [99]. Ongoing phase III studies are designed to further elucidate the role of anti-HER2 therapies in the treatment of CRC; nonetheless, routine assessment of HER2 is not recommended in the management of metastatic CRC.

### 2.6. PI3KCA

Phosphatidylinositol 3-kinase catalytic subunit alpha (PI3KCA) is a downstream effector of the EGFR as well as *RAS* and operates parallel to *BRAF* and has been identified in up to 25% of CRC [49,100]. Current evidence suggests that *PI3KCA* mutations render anti-EGFR mAb treatment of CRC ineffective with lower response rates and is associated with worse outcomes overall [101,102,103,104]. As the downregulation of COX2 is shown to inhibit PI3K signaling, an emerging potential therapeutic role has been suggested for aspirin [105]. Liao et al. demonstrated that patients with PI3KCA mutations who regularly took aspirin after CRC diagnosis had significantly longer cancer-specific survival at 5 years (HR 0.18, 95% CI 0.06–0.61, *p* < 0.001) and overall survival (HR 0.54, 95% CI 0.31–0.94, *p* = 0.01) compared to those who did not [105]. A phase II clinical trial was developed to investigate the use of MK-2206, a *Protein kinase B* (*AKT*)-inhibitor, in the treatment of *PI3KCA* mutant, *KRAS* wt mCRC, which had progressed on standard therapy; however, the trial was closed secondary to lack of accrual, with only one participant [106]. Further evidence is required to demonstrate a survival benefit for ASA use in PI3KCA mutant tumors or PI3KCA targeted drugs.

## 3. Novel Therapeutic Modalities

### 3.1. Oncolytic Viruses

Oncolytic viruses (OV) represent a promising approach to the treatment of various cancers. These viruses harness the body’s immune system and recombinant DNA technology to preferentially destroy malignant cells and enhance the anti-tumoral immune response. A variety of viral backbones (herpesvirus, adenovirus, reovirus, etc.) are engineered for tumor cell destruction, leading to multiple cycles of replication, destruction, re-infection, and continued lysis of cancer cells [107]. The most important clinical example of such viruses that have been engineered for oncolysis is talimogene laherparepvec, which was approved by the FDA for melanoma patients, demonstrating an increase in durable RR (16.3% durable RR with talimogene laherparepvec, compared to 2.1% durable RR with control) and overall RR (26.4% overall RR with talimogene laherparepvec, compared to 5.7% overall RR with control). It is under investigation in clinical trials for pancreatic cancer, breast cancer, and hepatocellular carcinoma [107,108]. Yang et al. demonstrated the efficacy of an oncolytic herpes simplex virus (HSV) in killing cultured colon cancer cells and colon cancer stem cells [109]. Furthermore, reovirus serotype 3 has been studied in a phase I trial of patients with mCRC and results suggest that patients mounted an adequate immune response resulting in robust lysis of mCRC cells [110]. Another phase I trial of reovirus serotype 3 has shown to be safe and well-tolerated with FOLFIRI co-treatment in *KRAS* mutant mCRC patients who had progression of disease on prior chemotherapy [111]. Ongoing trials continue to assess the safety and tolerability of various OVs in the treatment of CRC, but they have not yet succeeded to the advanced phases. Table 4 provides an overview of ongoing trials utilizing oncolytic viruses in the treatment of mCRC [112,113,114,115,116].

The major clinical challenge with OVs is often efficacy, not safety, and major morbidity and mortality from OV treatments are very rare [117]. Flu-like symptoms are the most common side effects of OV therapy [118]. Oncolytic virus toxicities are generally related to administration, and safety is increased with the administration of a virus with minimal shedding [117]. As an example, for herpes simplex viruses (HSV), latent infection is a concern, but strategies to circumvent this and increase safe use of this family for the development of oncolytic therapies have been described by Campadelli-Fiume et al. [119]. Moreover, the safety advantage for HSV is the availability of antivirals, such as acyclovir, to control unwarranted replication. The majority of OVs pass the phase I safety trials successfully. The major challenge in clinical translation of promising preclinical results has been suboptimal therapeutic efficacy and a lack of durable response. Different strategies are exploited to overcome these challenges and enhance therapeutic efficacy. These strategies include genetic modifications to improve tumor-specific entry targeting and increase oncotropism, enhance viral replication and spread, detarget normal native tissue, and post-entry targeting, and arming OVs with therapeutic genes [120]. The paradigm shift from OVs’ cytotoxic features to its immunomodulatory roles in recent years has brought OVs back into the cancer research spotlight [121]. These strategies are employed to enhance the immunomodulatory benefits of OVs. Preclinical findings support an enhanced immunoreactivity of tumors and synergistic tumor toxicity when OVs are combined with various immunotherapies including immune checkpoint inhibitors, which is considered to be a promising approach for the tumors that are not responsive to immune checkpoint inhibitors alone, mainly MSS tumors [122]. While OVs have failed to demonstrate meaningful therapeutic efficacy as a single agent, there are several potential advantages for combination therapy to achieve synergistic tumor-killing effects, which are the subject of several ongoing clinical trials.

### 3.2. CAR T-Cell

Chimeric antigen receptor T-cell (CAR-T) therapy represents another emerging technology in the treatment of cancer. This treatment modality utilizes modified CAR T-cells to recognize tumor-specific antigens and has been successfully used in the treatment of leukemia and lymphoma but has an increasingly promising role in the treatment of solid organ malignancies as well [123,124]. Zhang et al. demonstrated the safety of carcinoembryonic antigen (CEA) targeted CAR-T in patients with mCRC, notably 70% of their cohort had stable disease and 20% had improvement of disease burden as evidenced by cross-sectional imaging after treatment despite prior disease progression on standard therapy [125]. Hege et al. also demonstrated the safety and efficacy of tumor-associated glycoprotein (TAG)-72 targeted CAR-T therapy via systemic intravenous infusion versus hepatic artery infusion in patients with mCRC and hepatic CRC metastases, respectively [126]. The presence of the TAG-72 targeted CAR-T in the tumor was confirmed with tissue biopsy and serum TAG-72 levels were significantly decreased after CART treatment [126]. Similarly, Yang et al. conducted a phase I trial examining the use of CD133 targeted CAR-T in pancreatic adenocarcinoma, hepatocellular carcinoma and CRC [127]. Within this cohort of patients, 13% had partial remission, 61% had stable disease and there was a median PFS of 5 months after treatment [127]. Although the data support the safety of CAR-T in mCRC, subsequent clinical trials on this topic are warranted to evaluate the role of CAR-T as a more mainstream component of CRC treatment. Table 5 provides a summary of existing trials investigating CAR-T in mCRC treatment [128,129,130,131,132].

Despite all the advancements, CAR-T has not shown the same clinical benefit with solid tumors as it has with hematologic malignancies. Several distinct features of solid tumors lead to suboptimal clinical results with the current CAR-T cells. These challenges include the physical barrier, fibrosis and inflammation of the tumor microenvironment, inadequate T cell trafficking, and genetic heterogeneity with a lack of a universal tumor antigen to target [133,134]. To overcome these challenges, different strategies are being employed. As an example, the construction of co-stimulatory signals (e.g., CD28), expression of domain negative immunosuppressive factors such as IL-10 and TGF-beta, and transgenic cytokine expression (e.g., IL-12) have been considered to enhance immune response, improve trafficking, and reform the microenvironment and inhibit its immunosuppression [135]. In a similar strategy, as discussed with OVs, CAR-T can be engineered to elicit synergistic anti-tumor activity, in combination with conventional and novel therapeutic modalities [136]. Evidence supports enhanced immunotherapeutic efficacy with a combination of immune checkpoint inhibitors and CAR-T [137,138]. Clinical studies are being conducted to investigate them in CRC, including the subset resistant to standard immunotherapy. Moreover, they can be used as a platform to identify patient-specific neoantigens and the T cell subsets that are most effective in a personalized ex vivo approach.

## 4. Platforms for Personalized Approach

Cutting edge experimental ex vivo tumoroid platforms, including 3D cultures such as organoids and spheroids, and patient-derived xenografts, offer new and innovative approaches in the treatment of solid tumors such as colorectal cancer. Organoids are patient and/or stem-cell-derived tissue cultures that offer an ex vivo platform to investigate a plethora of diagnostic and therapeutic modalities [139,140]. These models have been shown to be a feasible pipeline for drug screening, predicting treatment response, mechanisms of treatment failure, and an overall viable option for investigating novel therapeutic modalities [141,142]. In a prospective study by van de Wetering et al., both normal and tumor tissue was extracted from CRC patients with active disease and the two organoids were compared to evaluate tumor-specific genetic mutations and identify drug sensitivities based on these mutations in a bid to better tailor therapeutic regimens [139]. Kim et al. do note that feasibility concerns may pose an obstacle to more mainstream incorporation of organoid technology in cases of rare genetic alterations; however, they also recognize the benefits of organoids in identifying potential gene-drug interactions; examples include an association between RNF43 mutant tumors and Wnt inhibitors [139,143]. Recent studies of colon and appendiceal cancers have demonstrated that organoids can be successfully established in up to 75% of cases and that organoid response to administered chemotherapeutic treatment is similar to that of response in parent tissue [144,145]. The feasibility of such studies offers hope for future applications of tumor organoids in a bid to drive clinical decision-making on an individualized basis.

Xenotransplantation represents another novel approach in the domain of precision medicine and involves the transplantation of patient-derived tumor tissue into animal models to assess in-vivo response to chemotherapeutic drugs [146]. Understandably, xenotransplantation is both costly and laborious, and it is often limited to more aggressive tumor types, which increases transplantation take rate [147]. Despite these limitations, xenotransplantation still offers an appealing in vivo route for studying clinical outcomes in regard to patient-specific tumors and they offer the opportunity to test newly developed treatment regimens [148].

As a bridge between organoids and xenotransplants, studies have been undertaken to create 3D tumor constructs with the advantage of more closely mimicking an in vivo model in a more controlled manner [147]. Proponents of this technology postulate that this approach allows for better control of the cellular environment than in xenotransplantation and allows for superior applicability than standard 2 dimentionsal (2D) organoids [147,149]. For example, with standard 2D organoid models, colorectal cancer specimens may adopt an epithelial structure that is not representative of the true phenotype in-vivo, leading to differences in signaling pathways, which may impact the effectiveness of tested treatments [149].

Overall, these platforms provide exciting new avenues to transform precision medicine into a personalized approach and explore the genetic heterogeneity of CRC for tangible outcomes in CRC therapies. While these technologies are not yet ready for prime time, clinical investigations are being conducted to evaluate their role in the optimization of the treatments and assessment of their efficacy in improving outcomes in a personalized approach. These efforts will transform the treatment paradigm from a conventional population-based treatment approach into a personalized approach based on the patient-derived platforms that provide predictive tools to guide and enhance treatment efficacy.

## 5. Conclusions and Future Directions

The precision approach, as a crucial component of modern systemic treatment, has substantially improved outcomes of metastatic colorectal cancer. Advancements in unraveling the molecular landscape of CRC and the involved signaling pathways have played an essential role in identifying therapeutic targets and developing novel targeted treatment modalities. Robustly designed clinical trials have successfully integrated molecular epidemiology and the precision approach with population-based evidence-derived treatment approaches. Precision approaches are, nonetheless, imperfect tools and failure is frequently observed. Molecular information and the presence of an actionable target do not always translate to effective targeting of a driver mutation. Significant genetic heterogeneity is a major obstacle for population-based approaches, and identification of a target, without the knowledge of intertwined confounding factors, is not sufficient to formulate an efficacious treatment approach accordingly. Moreover, developing resistance is the main reason for treatment failure, and there are no effective tools to predict or prevent it.

The personalized approach can conceptually overcome many of these inadequacies by identification, analysis, and ex vivo simulation of the treatment approach. While personalized technologies are evolving and are not widely adopted in clinical practice, they have enormous potential to fill the gaps of conventional and modern precision approaches. They also facilitate drug discovery and development of novel therapeutic modalities. Ongoing research is focused on optimizing and validating these technologies, which can revolutionize our management of advanced cancers. We will see groundbreaking changes in the design of future clinical trials by integrating personalized treatment approaches. It is anticipated that personalized treatments will eventually improve the therapeutic efficacy and outcomes and decrease treatment toxicity and financial burden by avoiding futile toxic and expensive treatments. That being said, personalized approaches have a long way to go to be integrated into clinical practice and are not quite ready for prime time, and thus, they represent an exciting avenue for cancer research with a meaningful impact on outcomes.

## Figures and Tables

**Table 1 cancers-12-03481-t001:** Common approved targeted therapy for colorectal cancer.

Signaling Pathway	Approved Targeted Therapy	Mechanism of Action	Indication
EGFR *RAS/RAF/ERK*	Cetuximab Panitumumab	anti-EGFR MoAb	RAS/BRAF wt mCRC (preferred 1st line for left-sided tumors)
MMR	Pembrolizumab Nivolumab	ICI-Anti PD-1 MoAb Anti PD-1 MoAb	dMMR/MSI-H mCRC
RAS/RAF/ERK	Encorafenib	BRAF kinase inhibitor	mCRC with BRAF V600E mutation
VEGF	Bevacizumab Regorafenib Ramucirumab Aflibercept	anti-VEGF MoAb anti-VEGF-R anti-IgG1 MoAb Recombinant fusion (VEGFR-1/2 & IgG1)	(Bev.) first-line regimen for unresectable metachronous mCRC; mCRC which has progressed with standard treatment
EGFR 2 (HER2)	Trastuzumab Lapatinib	Anti-HER2 MoAb TKI (anti-EGFR1/HER2)	HER2-overexpressed CRC after failing 1st line

Abbreviations: MoAb: monoclonal antibody; TKI: tyrosine kinase inhibitor; ICI: immune checkpoint inhibitor; mCRC: metastatic colorectal cancer; MMR: mismatch repair; BRAF: v-raf murine sarcoma viral oncogene homologB1; dMMR/MSI-H: deficienct mismatch repair, microsatellite instability high; VEGF: vascular endothelial growth factor; HER2: Human epidermal growth factor receptor 2; Bev.: bevacizumab.

**Table 2 cancers-12-03481-t002:** Recent trials involving checkpoint inhibitors in metastatic colorectal cancer (mCRC) treatment.

Study	NCT02060188 (Checkmate-142) *	NCT01876511 *	NCT02460198 (KEYNOTE-164) *
Phase	II	II	II
Population	74 recurrent or metastatic CRC patients with confirmed dMMR or MSI-H	41 patients with progressive mCRC regardless of MMR/MSI status or patients with other metastatic carcinoma known to have dMMR or MSI-H	124 mCRC patients with confirmed dMMR or MSI-H, cohort A was treated with ≥2 prior therapies, cohort B treated with ≥1 prior therapy
Treatment arms, prescribed regimen(s)	Nivolumab 3 mg/kg every 2 weeks	Pembrolizumab 10 mg/kg every 2 weeks	Pembrolizumab 200 mg every 3 weeks for up to 2 years
Primary endpoint(s)	Objective response rate (ORR)	ORR and PFS	ORR
Response rate, progression free survival and overall survival	ORR 68.9%, PFS 50% at 12 months, OS 73.4% at 12 months	ORR 40%, median PFS and median OS not reached at 12 months in dMMR/MSI-H mCRC patients	ORR 33% for both cohorts, median PFS was 2.3 months (cohort A) and 4.1 months (cohort B), median OS was 31.4 months (cohort A) and not reached at 27 months in cohort B
Grade 3 and 4 adverse events	Elevated lipase (8.1%), elevated amylase (2.7%)	Rash/pruritus (24%), hypothyroidism or other thyroid concerns (10%), pancreatitis (15%)	Grade 3–4 adverse events (16% cohort A, 13% cohort B) including colitis, hepatitis, pancreatitis, pneumonitis, skin reactions, arthralgias, asthenia and fatigue
Other significant findings	Patients with at least partial response had 12.8% greater decline in CEA level compared to patients with stable disease	There was a mean of 1782 somatic mutations per tumor in patients with dMMR or MSI-H	18% (cohort A) and 11% (cohort B) of participants had *RAS/BRAF* wt tumors

* ClinicalTrials.gov Identifier.

**Table 3 cancers-12-03481-t003:** Ongoing trials involving checkpoint inhibitors in mCRC treatment.

Study	NCT03202758 *	NCT04262687 *	NCT03832621 *
Phase	I/II	II	II
Condition/Disease/Population	Will include mCRC 48 patients with microsatellite stable (MSS), *RAS* mutant tumors. Actively recruiting and data not yet available	microsatellite stable mCRC patients with high immune infiltrate	Microsatellite stable, MGMT silenced mCRC
Intervention	Phase I: Durvalumab 750 mg every 2 weeks plus Tremelimumab 75 mg every 4 weeks plus FOLFOX during the 2 first cycles of treatment (1 month) Phase II: Durvalumab 750 mg every 2 weeks plus Tremelimumab 75 mg every 4 weeks plus FOLFOX	Drug: Capecitabine 2000 mg/m^2^/day, from day 1 to 14 of each cycle, Drug: Oxaliplatin 130 mg/m^2^ by IV infusion over 2 h, on day 1 of each cycle, Drug: Bevacizumab 7.5 mg/kg by IV infusion over 60 min, on day 1 of each cycle, Drug: Pembrolizumab 200 mg by IV infusion over 30 min, on day 1 of each cycle	Drug: Temozolomide 150 mg/sqm daily on days 1–5 every 4 weeks, Drug: Nivolumab 480 mg IV every 4 weeks, Drug: Ipilimumab (low-dose) 1 mg/Kg IV every 8 weeks
Endpoints	Safety (after Phase I) and Efficacy (after Phase II)	Assessing efficacy of pembrolizumab in combination with xelox and bevacizumab as first-line treatment for microsatellite stable (non MSI-H) mCRC	8-month PFS after treatment with combination of temozolomide, nivolumab and ipilimumab
Enrollment (total, open: yes/no)	48, Yes	55, Not yet	100, Yes

* ClinicalTrials.gov Identifier.

**Table 4 cancers-12-03481-t004:** Ongoing trials investigating oncolytic viruses in metastatic colorectal cancer.

Study	NCT04301011 *	NCT01394939	NCT01274624 *	NCT03206073 *	NCT03225989 *
Phase	I/II	I/II	I	I/II	I/II
Condition/Disease	Solid tumor, triple negative breast cancer, microsatellite stable colorectal cancer	Metastatic, refractory colorectal carcinoma	*KRAS* Mutant Metastatic Colorectal Cancer	Colorectal Cancer	Pancreatic adenocarcinoma; Ovarian cancer; Biliary carcinoma; Colorectal cancer
Intervention	Biological: TBio-6517, Pembrolizumab	Biological: JX-594, Irinotecan	Biological: REOLYSIN, Drug: Irinotecan, Drug: Leucovorin, Drug: Fluorouracil (5-FU), Drug: Bevacizumab	Biological: Pexa-Vec, Drug: Durvalumab, Drug: Tremelimumab	Drug: Load703 (oncolytic adenovirus serotype 5/35 encoding immunostimulatory transgenes: TMZ-CD40L and 41BBL)
Enrollment (total, open: yes/no)	84, not yet	52, Completed	36, Completed	35, Open	50, Open

* ClinicalTrials.gov Identifier.

**Table 5 cancers-12-03481-t005:** Ongoing trials investigating chimeric antigen receptor T-cells (CAR-T) in colorectal cancer.

Study	NCT03542799 *	NCT03692429 *	NCT03152435 *	NCT04348643 *	NCT03638206 *
Phase	I/II	I	I/II	I/II	I/II
Condition Disease	Metastatic colorectal cancer (mCRC)	Colorectal Cancer	EGFR-positive Colorectal Cancer	Solid tumor, lung cancer, colorectal cancer, liver cancer, pancreatic cancer, gastric cancer, breast cancer	Hematologic malignancies; HCC; Gastric Cancer; Pancreatic Cancer; Mesothelioma; Colorectal Cancer; Esophagus Cancer; Lung Cancer; Glioma; Melanoma; Synovial Sarcoma; Ovarian Cancer; Renal Carcinoma
Intervention	EGFR IL 12 CAR T-cells	Allogeneic CAR T-cells, CYAD-101	Biological: EGFR CAR T-cells	Biological: CEA CAR-T cells	Biological: CAR-T cell immunotherapy
Enrollment (total, open: yes/no)	20, not yet	36, yes	20, unknown	40, not yet	73, open

* ClinicalTrials.gov Identifier.

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
