# Peer review of "Precision Approaches in the Management of Colorectal Cancer: Current Evidence and Latest Advancements towards Individualizing the Treatment"

_cancers, 2020, doi:10.3390/cancers12113481_

Round 1

Reviewer 1 Report

This review does not live up to its title, which suggests that it will describe the latest advances in the shift to personalised treatment for colorectal cancer.  Unfortunately, the review is very generic and focusses too much on relatively old clinical data which has already been extensively discussed in the literature.  There is very little information on biomarkers, which are the fundamental drivers of personalisation.

Comments by section:

2.1 It is not necessary to describe all the phase III trials of cetuximab in detail.  Rather, it is far more important to describe biomarkers for EGFRi resistance.  Note that the much debated left vs right "biomarker" has not been prospectively confirmed. Should go into more detail re inhibitors of mutant KRAS eg alternatives to direct KRAS inhibition eg SHP2

2.2 Minor point - QUASAR was not recent.  It would be very interesting to dissect out the potential reasons for difficulty in applying checkpoint inhibitors to MSS tumours.  Also to comprehensively describe trials of novel approaches using checkpoint inhibitors for CRC including rationale eg IMblaze370.

2.3 BEACON is by far the most significant currently reported trial for V600E MT CRC.  What about non-V600E MT CRC?

2.4 The major problem with the VEGFi data is that there don't appear to be any clinically significant biomarkers. Why?

2.7 This section can be deleted

3.1 and 3.2 There is no discussion about which patients might benefit from these novel approaches and why.  On the face of it, CRC does not seem a suitable disease for these therapeutic modalities so why are they being pursued?

Reviewer 2 Report

I think the paper is interesting and evaluated old and new topic about target therapy in colorectal cancer.

I suggest to rewrite paragraph about EGF and VEGF as indicate in comments upload

Thanks

Reviewer 3 Report

Comments

  1. In this review authors provide a detailed review of the CRC approaches with the latest developments
  2. Simple summary:  Appropriately presented. Authors noted multiple pathways (EGFR, VEGF, MSI), however, they are abbreviated very early in the abstract. Either they have to elaborate or removed as the details are already provided in the manuscript at the later point.
  3. Introduction: Paragraph 1 is appropriately placed. It is suggested to add barriers to CRC screening in patients with and without insurance (PMID: 16116360, PMID: 30191078, PMID: 20409499). This calls for holistic attention with precision, personalization based on the patient's needs. This puts the article more into perspective.
  4. The second and third paragraph of the introduction is well written
  5. Authors are strongly recommended to provide a flow sheet/ table of the traditional (RAS, EGFR, MSI, MMR, BRAF, VEGF) and novel therapeutic targets (viruses, CAR T-cell). 
  6. Section 2.1 first paragraph is appropriately written. The second paragraph is abnormally long and needs to split. Alternately a table comparing CAPRI, CRYSTAL, BOND, TAILOR trial will achieve brevity, and messages can be provided across easily.
  7. MSI and immune checkpoint inhibitors: Authors note in great detail about their pathways in CRC and drug targets. Using the same, they are encouraged to discuss adverse events with these agents (immune-mediated inflammation, etc). This gives a holistic idea about the pitfalls of these therapeutics. 
  8. Similar aspects hold good for BRAF, VEGF, HER2.
  9. Authors appropriately presented novel targets in concise yet up to the point fashion
  10. Please add a feature of “Future research and Gaps” which will provide further information about the aspects which need attention in the future.

Round 2

Reviewer 1 Report

I thank the authors for their responses.  Unfortunately, they have not addressed my first and most important comment, which remains the major criticism of the review.  i have pasted it below:

This review does not live up to its title, which suggests that it will describe the latest advances in the shift to personalised treatment for colorectal cancer.  Unfortunately, the review is very generic and focusses too much on relatively old clinical data which has already been extensively discussed in the literature.  There is very little information on biomarkers, which are the fundamental drivers of personalisation.

Author Response

  • This review does not live up to its title, which suggests that it will describe the latest advances in the shift to personalised treatment for colorectal cancer.  Unfortunately, the review is very generic and focusses too much on relatively old clinical data which has already been extensively discussed in the literature.  There is very little information on biomarkers, which are the fundamental drivers of personalization
    • Thank you for your comment. We divided this manuscript into two parts. First is the important involved pathways and ongoing or past evidence of the precision approaches derived from population-based evidence. Next, we discussed the emerging technology that is now increasingly being used to individualize the treatments. Herein, we discussed the platforms through which individualized approaches can be implemented. We have discussed the role of bioengineering the platforms to attack tumor-specific targets. We have also discussed individualized approaches to tailor immunotherapeutics at the personalized level. Discussing biomarkers is also a different interesting angle. In our opinion, biomarkers have two different implications. One is the extension of precision approaches which we extensively discussed the ones pertinent to colorectal cancer, and the other is their application as targets for various emerging technologies that we discussed. However, in a nutshell, we discussed that the pharmacogenetics information that is obtained from these platforms can guide the therapy, but we felt a list of specific biomarkers (not necessarily specific to colorectal cancer) would require a separate manuscript.

Reviewer 2 Report

see in file attached

Author Response

  • Correct : ‘administration of anti-EGFR based and immunotherapy respectively’ (INSTEAD OF ‘antiVEGF therapy)
    • This stylistic change was made “RAS mutations and deficiencies in the mismatch repair pathway guide decisions regarding administration of anti-EGFR based and immunotherapy, respectively”

  • Delete lines 95-99 From ‘a plethora ..’ to ..’..done (23)’ (and delete reference 23 consequently) If you want to mention CAPRI study move this sentence after line 146
    • Sentence regarding CAPRI trial moved after line 146 “Furthermore, findings of the CAPRI trial, demonstrated improved progression-free survival (PFS) in RAS wt mCRC patients treated with anti-EGFR mAb by almost 2 months (PFS of 6.4 months in patients who were treated with Cetuximab and FOLFOX compared to 4.5 months with FOLFOX alone) [34].”

  • Delete lines 107-110 (from ‘in the preclinical setting ..’ to ‘.. KRAS mutant CRC tumors’) and delete consequently references 27-28
    • These lines were deleted

  • Line 100: Add after ‘mCRC’ ..’ in human colorectal xenograft’
    • “Multiple studies have examined these agents in use with Irinotecan-based chemotherapy regimens, as anti-EGFR mAb were postulated to provide benefit to Irinotecan-resistant mCRC in human colorectal xenograft [23]

  • Move lines 101-107 (‘the phase II OPUS …OS 23.9 months versus 19.7 months p017 (26)) after ‘oxaliplatin based regimens’ (line 127)
    • These lines were moved

  • Line 130 add after …to FOLFOX alone : ‘ after a post hoc evaluation with extending KRAS mutations’
    • The following was added “In regards to targeted anti-EGFR treatment in use with Oxaliplatin-based regimens, a phase III trial carried out by Douillard and colleagues showed improved PFS (PFS increased by 2.2 months with the addition of panitamumab; 10.1 months versus 7.9 months) and OS (OS increased by 5.8 months with the addition of panitumumab; 26 months compared to 20.2 months) with FOLFOX and Panitumumab in comparison to FOLFOX alone after a post hoc evaluation with extending KRAS mutations [27].”

  • Line 111 after ‘a phase III trial add that compared FOLFIRI plus Cetuximab versus FOLFIRI in first line treatment mCRC’…this sentence: ‘the post hoc evaluation of the study extending RAS mutation showed…’
    • The following was added “The post hoc evaluation of the study extending RAS mutation showed that there may be benefit to addition of cetuximab if RAS mutation signals were less than 5% [24].”

  • Line 146 add: ‘potential bias of this study is that only KRAS wt (and no all RAS wt) patients were I included.’
    • The following was added “One potential bias of this study is that only KRAS wt (and no all RAS wt) patients were included.”

  • Line 162 correct PFS instead of PRS
    • Corrected

  • Line 165 delete.. ‘of left sided lesions’ (is pleonastic)
    • Deleted

  • Line 173 add ‘compared to bevacizumab’ (instead of or bevacizumab)
    • Changed “Interestingly, neither PFS nor OS were improved with addition of cetuximab compared to bevacizumab without controlling for KRAS wt status [43].”

  • Line 227 ADD immune related PFS at 20 weeks of 78% (versus 11%)
    • Added “Le and colleagues found that treatment with pembrolizumab in patients with dMMR resulted in a RR of 40% (versus 0%) and immune related PFS of 78% (versus 11%) in comparison to those without MMR deficiency [55].”

2.4 VEGF

  • Line 315 delete ‘adjuvant’
    • deleted
  • Line 362 ADD before ref 94 ‘in KRAS wt patients’
    • Added “A similar study by Venook and colleagues also studied addition of cetuximab versus bevacizumab to chemotherapy, and while there was a trend towards improved OS with cetuximab, this finding was not statistically significant (30.0 months versus 29.0 months p=0.08) in KRAS wt patients [42]”

  • Delete reference 21 Add a number before reference ‘Allegra CJ…(after ref 19)
    • Reference 21 deleted and number added before Allegra CJ reference

  • Delete reference 52 and ADD 52 to the reference starting with ‘ Sargent DJ…’
    • Reference 52 deleted and added number to Sargent DJ reference